# Phytoestrogen Coumestrol Selectively Inhibits Monoamine Oxidase-A and Amyloid β Self-Aggregation

**DOI:** 10.3390/nu14183822

**Published:** 2022-09-16

**Authors:** Su Hui Seong, Bo-Ram Kim, Myoung Lae Cho, Tae-Su Kim, Sua Im, Seahee Han, Jin-Woo Jeong, Hyun Ah Jung, Jae Sue Choi

**Affiliations:** 1Division of Natural Products Research, Honam National Institute of Biological Resource, Mokpo 58762, Korea; 2Division of Botany, Honam National Institute of Biological Resource, Mokpo 58762, Korea; 3Department of Food Science and Human Nutrition, Jeonbuk National University, Jeonju 54896, Korea; 4Department of Food and Life Science, Pukyong National University, Busan 48513, Korea

**Keywords:** *Pueraria lobata*, coumestrol, monoamine oxidase-A, Aβ aggregation

## Abstract

*Pueraria lobata* leaves contain a variety of phytoestrogens, including flavonoids, isoflavonoids, and coumestan derivatives. In this study, we aimed to identify the active ingredients of *P. lobata* leaves and to elucidate their function in monoamine oxidase (MAO) activation and Aβ self-aggregation using in vitro and in silico approaches. To the best of our knowledge, this is the first study to elucidate coumestrol as a selective and competitive MAO-A inhibitor. We identified that coumestrol, a coumestan-derivative, exhibited a selective inhibitory effect against MAO-A (IC_50_ = 1.99 ± 0.68 µM), a key target protein for depression. In a kinetics analysis with 0.5 µg MAO-A, 40–160 µM substrate, and 25 °C reaction conditions, coumestrol acts as a competitive MAO-A inhibitor with an inhibition constant of 1.32 µM. During an in silico molecular docking analysis, coumestrol formed hydrogen bonds with FAD and pi–pi bonds with hydrophobic residues at the active site of the enzyme. Moreover, based on thioflavin-T-based fluorometric assays, we elucidated that coumestrol effectively prevented self-aggregation of amyloid beta (Aβ), which induces an inflammatory response in the central nervous system (CNS) and is a major cause of Alzheimer’s disease (AD). Therefore, coumestrol could be used as a CNS drug to prevent diseases such as depression and AD by the inhibition of MAO-A and Aβ self-aggregation.

## 1. Introduction

Monoamine oxidase (MAO) is a group of flavoenzymes that catalyze the oxidation of biogenic and xenobiotic amines including primary (dopamine, serotonin, norepinephrine, etc.), secondary (adrenaline), and tertiary amines, producing H_2_O_2_ and aldehyde. Thus, MAOs play a key role in regulating monoamine levels in the central nervous system (CNS). Of the two isomers (A and B types) of MAO, MAO-A exhibits substrate selectivity for serotonin, melatonin, norepinephrine, and epinephrine, and MAO-B exhibits substrate selectivity for phenylethylamine and benzylamine. However, both isomers are known to non-selectively catalyze dopamine and tyramine [1]. Therefore, MAO-A is widely used as a drug target for anxiety disorders such as depression and MAO-B for neurodegenerative diseases such as Parkinson’s disease (PD) [2]. Unfortunately, MAO inhibitors (MAOIs) are not popular as a CNS drug candidate due to side effects such as hypertension, which occurs because of the ‘cheese effect’ caused by irreversible and nonselective MAOIs. However, selective and reversible MAOIs (moclobemide) have also been developed, which do not cause the ‘cheese effect,’ and therefore, their administration does not require dietary restrictions [3]. In addition, MAO-B levels were found to be increased in the brain of AD patients, and MAO-B activation increased amyloid beta (Aβ) production through gamma-secretase activation [4]. Aggregated Aβ is a major component of senile plaques, which is the key cause of AD and has been widely studied as a biomarker in the development of therapeutic drugs for AD [5]. Thus, it is essential to develop a lead compound that affects MAO activation and Aβ production/aggregation. 

*Pueraria lobata* (Willd.) Ohwi belongs to the Leguminosae family and is one of the climbing plants that have been used as food or traditional medicine in many countries, including Korea, China, and Japan [6]. Due to the various pharmacological properties of *P. lobata*, it has been widely used in folk remedies or herbal medicines since ancient times; research on its active ingredients and pharmacological activity is in progress. The representative components of the roots and leaves of *P. lobata* are puerarin (daidzein-8-*C*-glucoside) and robinin (kaempferol 3-*O*-robinoside-7-*O*-rhamnoside), respectively. In addition, *P. lobata* contains many isoflavones, flavonoids, triterpenoids, and coumestan derivatives [6,7]. In particular, *P. lobata* is rich in isoflavones that help relieve female menopausal symptoms; hence, it is widely used by women. The isoflavone-rich fraction of *P. lobata* exerts neuroprotective effects by enhancing the metabolism of neurotransmitters, and puerarin exerts anti-PD effects by protecting dopaminergic neurons [8,9]. In addition, the ethanol extract of *P. lobata* roots exhibited anti-depressant activity in a cerebral ischemia reperfusion mouse model [10]. However, the systematic studies on the MAO inhibitory effects of *P. lobata* leaves and its component are still limited. The discovery of an active ingredient from *P. lobata*, which can be easily consumed as food, is of great nutritional and industrial importance. Therefore, in this study, we aimed to identify the active ingredients of *P. lobata* leaves and to elucidate their function in MAO activation and Aβ aggregation using the analytical technique HPLC–Q-TOF–MS, in vitro enzyme inhibitory and protein aggregation assays, and in silico computational approaches. In silico studies are useful to understand the role of the steric hindrance of the functional groups of phytochemicals in the active site of a target enzyme. In addition, a kinetic assay, which observes changes in enzyme activity according to the substrate and inhibitor (=active ingredient) concentration, was conducted to identify the inhibitory type of active ingredient. To further support their potential as a drug, the pharmacokinetic parameters of the active molecules were also predicted.

## 2. Materials and Methods

### 2.1. Chemicals

Human monoamine oxidase (hMAO) isozymes, robinin, rutin, nicotiflorin, daidzin, genistin, coumestrol, daidzein, genistein, curcumin, amyloid β peptide 25–35 (Aβ_25–35_), hexafluoroisopropanol (HFIP), and selegiline hydrochloride were purchased from Sigma-Aldrich (St. Louis, MO, USA). All solvents used for high pressure liquid chromatography coupled to quadrupole time-of-flight mass spectrometry (HPLC–Q-TOF–MS) were liquid chromatography–mass spectrometry grade and purchased from Merck (Darmstadt, Germany).

### 2.2. Plant Source

*P. lobata* leaves were collected from Geogeum-do islands (Goheung-gun, Jeollanam-do, Korea) in August 2021. The plant was authenticated by Dr. S. Han of Honam National Institute of Biological Resource (voucher no. shan2021-142). The 70% ethanol extract of *P. lobata* leaves (PL-L-70E) was obtained from the Bank of Bioresources from Island and Coast (BOBIC), Republic of Korea (registered no. HNIBR NP374).

### 2.3. HPLC–ESI-Q–TOF–MS Analysis

The PL-L-70E was separated on the Ultimate™ 3000 UHPLC system using the Hypersil GOLD™ C_18_ column (Cat. 25005-254630), equipped with the Chromeleon Software 7.3 (ThermoScientific, Pittsburgh, PA, USA), for component analysis. The mobile phase consisted of a gradient of 0.1% HCOOH (A) and acetonitrile containing 0.1% HCOOH (B) as follows: 0–60 min, B 10% to 45%; 60–60.1 min, B 45% to 100%; and 60.1–65 min, B 100%. The sample injection volume was 2.5 μL, column temperature was 25 °C, and flow rate was 0.5 mL/min. MS analysis was performed using Xevo G2-XS Q-TOF–MS with MassLynx v10 (Waters Corporation, Milford, MA, USA). ESI–MS^e^ mode was used to obtain low- and high-resolution spectral data in the negative mode. MS conditions were as follows: acquisition mass range, 50 to 1500 Da with a 0.5 s scan time; cone voltage, 40.0 V; source and desolvation temperatures, 100 °C and 350 °C, respectively. Mass calibration was performed using 0.5 mM sodium formate. The mass was corrected during acquisition using 200 pg/mL leucine enkephaline.

### 2.4. In Vitro hMAO Inhibitory Assay 

The inhibitory potential of the *P. lobata*-derived phytochemicals against MAO was evaluated through the luminometric method. Various doses of phytoestrogens in 0.1 M HEPES buffer (pH 7.5) with 5% 1,2,3-propanetriol for MAO-A and 5% 1,2,3-propanetriol and 10% dimethyl sulfoxide for MAO-B were mixed with the substrate (160 µM for MAO-A and 16 µM for MAO-B) and 0.5 µg of the enzyme and incubated at 25 °C. After 1 h incubation, reconstituted luciferin-detection reagent was added to the reaction mixture and incubated at 25 °C for 20 min. After that, luminescent signals were monitored using a microplate reader (Biotek, Winooski, VT, USA). Selegiline hydrochloride (final concentrations: 6.25 to 25 µM) was used as a positive control [11].

### 2.5. Kinetic Assay for hMAO-A

The type of inhibition exhibited by coumestrol was analyzed based on the changes in enzyme activity according to various concentrations of the substrate and the inhibitor [12]. The experimental method was the same as that used for the hMAO inhibitory assay; 0.5 µg of enzyme, 40–160 µM substrate and 0, 2.5, 10, and 20 µM coumestrol were used. A Lineweaver–Burk plot (LB plot) was generated using SigmaPlot 12.0 based on the results of the kinetic assay and secondary plots of the LB plot (LB-2nd plot) were generated using the exploratory EK macro module (Appendix A). 

The equation for the LB plot is as follows:(1)$$\frac{1}{V}=\frac{K_{m}+\left[S\right]}{V_{max}\left[S\right]}$$

The equations for the LB-2nd plots are as follows:(2)$$\frac{K_{m,app}}{V_{max,app}}=\frac{K_{m}}{V_{max}}(1+\frac{\left[I\right]}{K_{ic}})$$
(3)$$\frac{1}{V_{max,app}}=\frac{1}{V_{max}}(1+\frac{\left[I\right]}{K_{iu}})$$

### 2.6. Self Aβ_25–35_ Aggregation Assay

The Aβ_25–35_ self-aggregation assay was performed following the method described by Naldi et al. [13]. First, peptide was pretreated with HFIP for 1 day at 22 °C to obtain non-amyloidogenic conformation. Various doses of phytoestrogens in 34.5 mM phosphate buffer (pH 7.4) with 17.5% MeCN were mixed with a 0.1 mM monomeric Aβ_25–35_ solution in a 1:29 ratio (*v*/*v*) and incubated overnight at 4 °C. After incubation, the reaction mixture was supplemented with 0.025 mM thioflavin-T in 50 mM glycine–NaOH buffer (pH 8.5). The fluorescence emission was monitored at 490 nm with excitation at 446 nm using the Gemini XPS (Molecular Devices, Sunnyvale, CA, USA). Curcumin was used as the standard [13].

### 2.7. In Silico Docking Simulation

The X-ray crystallographic structure of hMAO-A was obtained from the PDB (ID 2z5x) [14]. Water and co-ligands from the structure were eliminated using the Discovery Studio (v17.2, Accelrys, San Diego, CA, USA). The 3D structure of coumestrol was generated using Marvin Sketch (v17.1.30, ChemAxon, Budapest, Hungary). Fifteen docking postures were generated with the same parameters using AutoDock 4.2 [15]. The results were scrutinized using Discovery Studio v17.2.

### 2.8. Prediction of Pharmacokinetic Parameters

The pharmacokinetic characteristics of coumestrol were calculated using Marvin Sketch (v17.1.30) and the PreADMET server v1.0 (https://preadmet.bmdrc.kr/, accessed on 1 June 2022).

### 2.9. Statistical Analysis

The 50% inhibitory concentration (IC_50_) calculated from the dose–inhibition curve is expressed as the mean ± standard deviation (SD) of three independent experiments. The statistical significance of the group treated with the tested compounds against the control group for the Aβ_25–35_ self-aggregation assay was calculated via Student’s *t*-test (Microsoft Excel 2019, Microsoft Corporation, Seattle, WA, USA).

## 3. Results

### 3.1. Extracted Ion Chromatogram (EIC) of Compounds Present in the PL-L-70E

The chemical constituents of the PL-L-70E were analyzed using HPLC–Q-TOF–MS^e^ (Figure 1 and Table 1). We determined that various glycosides of quercetin and kaempferol were present in *P. lobata* leaves based on the comparison of their retention times (RT) with those of standard compounds and the analysis of their fragmentation patterns. In particular, robinin was identified as the most abundant component. In addition, flavan-3-ols (quercetin-3-*O*-robinobioside, rutin, kaempferol-3-*O*-robinobioside, and nicotiflorin), isoflavones (daidzein, genistein, daidzin, and genistin), and the coumestan-derivative coumestrol were also present in this extract.

### 3.2. Inhibitory Activity of PL-L-70E and Its Constituents against Human MAO Isozymes

The protocol for the MAO assay was verified using selegiline hydrochloride as a positive control. In addition, since factors such as organic solvents, metal ions (such as Cu^2+^, Cd^2+^, and Al^3+^), pH, and temperature affect MAO activity [16,17,18], the assay was conducted in a well-controlled laboratory environment. The optimal conditions for the enzymatic reaction, including reaction time, temperature, pH, and the concentrations of the enzyme and substrate, were established based on the previous study [19]. All the test samples were diluted in a reaction buffer containing 10% or less DMSO, and the solvent-control group was tested together for each assay to minimize false-positive/false-negative effects.

The PL-L-70E exhibited dose-dependent inhibitory activity against MAO isozymes (Figure 2A), with high selectivity for MAO-A compared to that for MAO-B. Therefore, the MAO inhibitory efficacy of these constituents was further evaluated. Among the tested compounds, coumestrol was the most effective MAO-A inhibitor, with an IC_50_ of 1.99 μM. However, this compound showed moderate inhibition against MAO-B, with an IC_50_ of 77.79 (selective index (SI) = 0.02). As shown in Table 2, genistein also showed good inhibitory activity against both MAO isozymes with IC_50_ values of 4.77 and 3.42 μM for MAO-A and -B, respectively. On the other hand, daidzein, which has a structure similar to that of genistein without a hydroxyl group at the C5 position, showed a weak inhibitory effect against both MAO isozymes (IC_50_ values of 304.05 and 356.87 μM for MAO-A and -B, respectively). Previously, Zarmouh et al. [20] also demonstrated that genistein exhibits excellent inhibitory activity against MAOs through an enzyme kinetic study and docking simulations, and these results are consistent with our present results. In addition, rutin showed weak MAO-A inhibitory activity, but others showed no activity against both MAO isozymes under the tested concentration. 

### 3.3. Competitive Inhibition of hMAO-A by Coumestrol 

To the best of our knowledge, for the first time, the potent and selective inhibitory activity of coumestrol against MAO-A was elucidated. Therefore, an enzyme kinetic analysis was performed to elucidate the mode of inhibition of MAO-A by coumestrol. As in the previous studies, it was confirmed that the lower the concentration of the substrate, the lower the enzyme reaction rate of the control group (treated with only buffer) [19]. 

As shown in Figure 2B,D, the y-intercept of each linear regression did not change even when the substrate and coumestrol concentrations were changed, and this kinetic pattern is that of a typical competitive inhibitor. According to the secondary plots (Figure 2C,D), the inhibition constant of coumestrol for MAO-A was calculated as 1.32 µM (Table 3). Furthermore, in silico docking simulations were performed to predict the binding sites of coumestrol to MAO-A and key residues that affect its binding. As shown in Figure 2E,F, coumestrol was stably docked to a site known as the major active site of MAO-A [14]. The ketone moiety of coumestrol interacted with nitrogen at position 5 of the isoalloxazine ring of FAD via a hydrogen bond (2.16 Å), and the hydroxyl moiety of the B-ring of coumestrol and the Tyr444 residue of MAO-A formed a hydrogen bond (1.85 Å). Coumestrol also formed a pi–pi bond with Phe352 and Tyr407. Thus, these strong hydrogen bonds and pi–pi interactions may have affected the binding ability of coumestrol to MAO-A.

### 3.4. Inhibitory Activity of Phytoestrogens against Aβ_25–35_ Self-Aggregation

Four phytoestrogens that exhibited inhibitory activity against MAO and curcumin (a well-known inhibitor of Aβ_25–35/1–42_ self-aggregation) were investigated to evaluate their inhibitory potential against Aβ_25–35_ self-aggregation based on the fluorescence of thioflavin-T (Appendix A). The optimal conditions for the Aβ_25–35_ aggregation, including the ratio of Aβ_25–35_ and inhibitor, temperature, reaction time, and pH, were established based on the previous study [13]. Figure 3A shows the inhibitory activity (%) of the four phytoestrogens and curcumin against self Aβ_25–35_ aggregation after 24 h of incubation. The extent of Aβ_25–35_ self-aggregation decreased by 76.14% when co-treated with 100 μM coumestrol compared to that of the control group. Furthermore, 100 μM rutin, daidzein, and genistein reduced Aβ_25–35_ self-aggregation by 49.31, 35.54, and 34.90%, respectively. As shown in Figure 3B, coumestrol inhibited Aβ_25–35_ aggregation in a dose-dependent manner with an IC_50_ value of 37.40 ± 1.70 μM, whereas the curcumin had an IC_50_ value of 10.57 ± 1.31 µM. 

### 3.5. Pharmacokinetic Parameters of Coumestrol

Based on the chemical structure of coumestrol, the pharmacokinetic parameters were predicted. The pharmacokinetic analysis of coumestrol revealed that its blood–brain barrier (BBB) permeability score was 0.76, which indicated that coumestrol could pass across the BBB with moderate absorption by the CNS. In addition, coumestrol showed 93.51% human intestinal absorption (HIA), which implies that it can be easily absorbed by the human intestine and is suitable for oral delivery. According to the CMC-like rule [21], the lipophilicity index of drugs, the log P for neuronal drugs should be between 1.3 and 4.1. As shown in Table 4, the log P_o/w_ value of coumestrol was 2.94. Coumestrol was also predicted to be a non-inhibitor of p-glycoprotein. Based on the analysis of its mutagenic properties, coumestrol was predicted to be non-carcinogenic to rats and mice. The predicted CNS MPO score suggested that coumestrol has suitable CNS-drug-like properties based on several physicochemical properties (partition coefficients, the number of hydrogen bond donors, pKa, molecular size, and topological polar surface area values) [22]. Therefore, these predicted pharmacokinetic properties project coumestrol as an attractive CNS drug candidate for the inhibition of MAO and the self-assembly of the Aβ peptide.

## 4. Discussion

In this study, we investigated the inhibition of MAO by the PL-L-70E and its constituents. We determined that coumestrol showed the best inhibitory effect against MAO-A, followed by genistein, daidzein, and rutin. In the case of MAO-B, genistein exhibited the strongest inhibition, followed by coumestrol and daidzein. However, the remaining compounds did not exhibit a significant inhibitory effect against either isozyme. Moreover, for the first time, we discovered that coumestrol acts as a selective and competitive MAO-A inhibitor.

Coumestrol is a coumestan-derivative and is found in *P. lobata*, soybean, alfalfa (*Medicago sativa*), and *Trifolium* sp. [23,24]. Coumestrol is a phytoestrogen with antioxidant, anti-inflammatory, and estrogenic activities [25,26]; it is biosynthesized from daidzein, a representative isoflavone that is abundant in the Leguminosae family [24]. Zarmouh et al. reported that psoralidin derived from *Psoralea corylifolia* seeds, which has an additional prenyl moiety at the C2 position of coumestrol, did not show a significant inhibitory effect against either MAO-A or -B [27]. In another study, it was confirmed that glycyrol derived from *Glycyrrhiza uralensis* roots, which has an additional prenyl group at the C2 position and a methoxyl group at the C3 position of coumestrol, had an IC_50_ value of 29.5 μM against MAO-B (no activity below 40 μM for MAO-A) [28]. Unlike psoralidin and glycyrol, in the present study, coumestrol exhibited potent and selective inhibitory effect against MAO-A at low concentrations (SI = 0.02). Presumably, the functional groups attached to C2 and C3 of psoralidin and glycerol form unnecessary interactions with other amino acid residues of MAO-A, preventing the ligand from docking to the active site of the enzyme, thereby lowering the inhibitory activity of the enzyme. However, an in silico docking simulation analysis demonstrated that coumestrol was stably docked to the substrate-binding site by forming hydrophobic or hydrogen bonds with the active-site residues (Tyr444, Tyr407, and Phe352) of MAO-A and its co-factor FAD. In addition, it was confirmed that coumestrol interacts with Ile335, which represents the major structural difference between MAO isoforms and affects the selectivity for MAO-A through a hydrophobic bond (pi–alkyl) [29,30]. Furthermore, the enzyme kinetics analysis according to different coumestrol and substrate concentrations revealed that coumestrol acts as a competitive inhibitor. The Lineweaver–Burk plot and its secondary plots revealed that V_max_ remained constant despite changes in substrate or coumestrol concentrations, which corresponds to a typical competitive inhibition pattern. Therefore, when coumestrol is consumed, there could be no side effects (ex. hypertension) linked to the ‘cheese effect’ caused by the consumption of nonselective MAOIs. However, an additional reversibility assay is needed to confirm whether coumestrol is a reversible inhibitor. In addition, it is needed to check the thermodynamic effect of the inhibitor on the enzyme through a scan of the changes in thermal stability in the free enzyme and inhibitor–enzyme complex.

The thioflavin-T assay for inhibitory compounds (rutin, daidzein, genistein, and coumestrol) against MAO was used to investigate their inhibitory effect against the self-aggregation of Aβ, which is known to cause neuroinflammation and AD. Coumestrol exhibited the most potent inhibitory effects with an IC_50_ of 37.40 μM. In a previous study, it was reported that a nanomolar concentration of coumestrol exhibited neuroprotective effects in mouse astrocytes by significantly reducing Aβ-induced cytotoxicity and inflammatory cytokine levels [31]. In addition, coumestrol also exhibited an in vitro inhibitory effect against β-secretase, which affects the formation of Aβ [32]. Thus, the inhibitory action of coumestrol against Aβ self-aggregation and β-secretase may cause the Aβ peptide to remain in the soluble monomer state, thereby facilitating the clearance of the peptide from the brain through normal physiologic mechanisms [33]. 

In this study, the potential of coumestrol as a CNS drug for the treatment of AD and depression was confirmed via in vitro and in silico approaches, and its pharmacokinetic properties were also predicted to support its potential. Coumestrol was predicted to possess suitable lipophilicity as a neuronal drug and to have normal BBB permeability [21]. Moreover, we proved that the structure of coumestrol is suitable for its development into a CNS drug based on the CNS MPO score, considering various common physicochemical properties [22]. In several studies, Aβ_1–40/1–42_ reduced the expression of p-glycoprotein in the BBB, thereby interrupting the clearance of Aβ in the brain [34,35]. Therefore, BBB permeable coumestrol could be a potential therapeutic drug that acts by inhibiting Aβ aggregation and improving Aβ clearance in the brain.

## 5. Conclusions

The phytoestrogen coumestrol, present in *P. lobata* leaves, acts as a selective and competitive MAO-A inhibitor by competing with the substrate at the substrate-binding site and stably binding to key residues and FAD in the active site of the enzyme. In addition, coumestrol significantly reduced the self-aggregation of Aβ, which is known to be a major cause of AD. Taken together, this study suggests that coumestrol may act as a food-derived natural anti-depressant and anti-AD material that can treat or prevent diseases related to MAO-A activation and Aβ self-aggregation.

## Figures and Tables

**Figure 1 nutrients-14-03822-f001:**
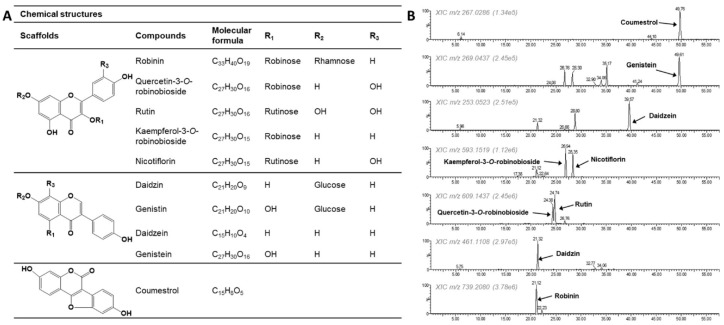
Chemical structures (**A**) and extracted ion chromatograms (**B**) of compounds present in PL-L-70E.

**Figure 2 nutrients-14-03822-f002:**
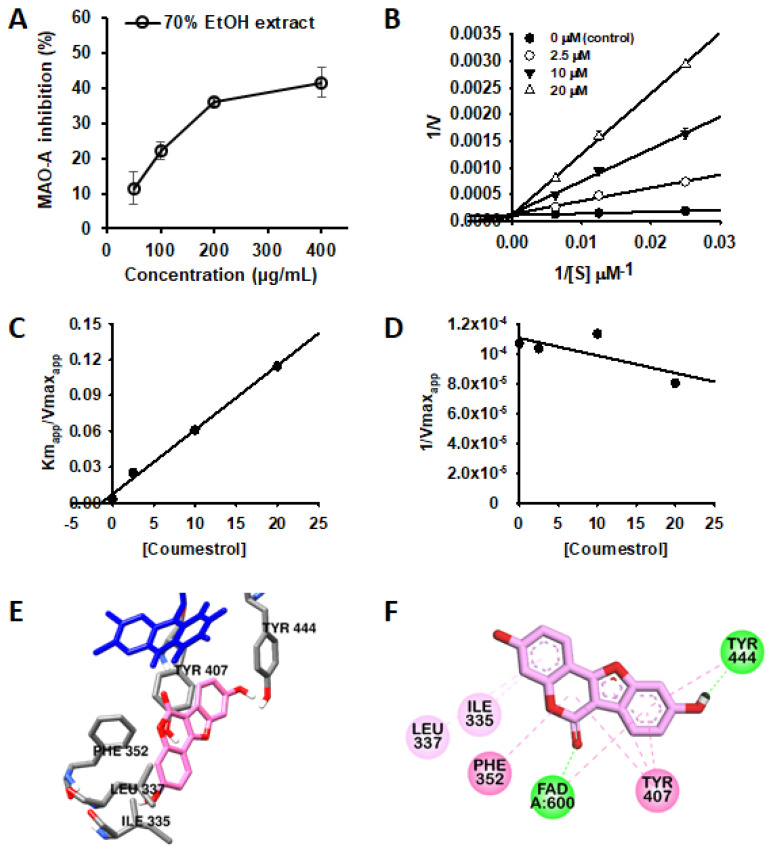
Dose-dependent inhibition of MAO-A by PL-L-70E (**A**). Lineweaver-Burk plots (**B**) and secondary plots (**C**,**D**) of coumestrol for hMAO-A inhibition. Data are presented as mean ± SD (*n* = 3). Comparative binding orientation (**E**) and 2D interaction (**F**) of coumestrol (pink stick) at the catalytic site of hMAO-A. FAD and residues of the enzyme are represented as blue and gray sticks, respectively.

**Figure 3 nutrients-14-03822-f003:**
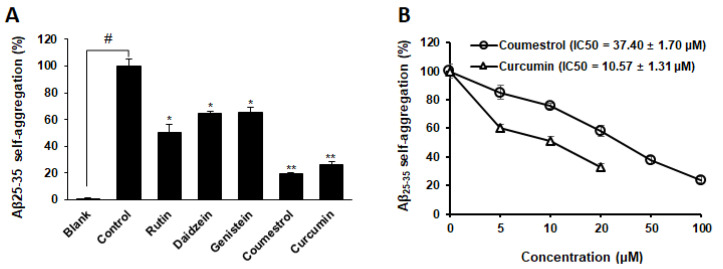
Extent of Aβ_25–35_ self-aggregation in the presence and absence of phytoestrogens (100 μM) along with positive control (20 μM curcumin) based on a quantitative thioflavin-T binding assay (**A**). Data are shown as mean ± SD (*n* = 3). ^#^
*p* < 0.0001 denotes a substantial difference from the non-aggregated group (blank). * *p* < 0.001 and ** *p* < 0.0001 denote substantial differences from the aggregated Aβ_25-35_ (100 μM) group (control). Dose-dependent inhibition of Aβ_25–35_ self-aggregation by coumestrol (**B**).

**Table 1 nutrients-14-03822-t001:** Identification of phytochemicals present in the PL-L-70E by HPLC–Q-TOF–MS^e^ in negative ion mode.

Compounds	RT (min)	Measured Mass	Molecular Formula	Error (ppm)	Fragment Ions (*m*/*z*) ^a^
Robinin	21.12	739.2080 [M-H]^−^	C_33_H_40_O_19_	−8.0	593.1519
Quercetin-3-*O*-robinobioside	24.38	609.1437 [M-H]^−^	C_27_H_30_O_16_	−3.1	301.0340, 300.0283
Rutin	24.74	609.1437 [M-H]^−^	C_27_H_30_O_16_	−3.1	301.0340, 300.0283
Kaempferol-3-*O*-robinobioside	26.94	593.1519 [M-H]^−^	C_27_H_30_O_15_	2.2	285.0408, 284.0315, 269.0437
Nicotiflorin	28.35	593.1519 [M-H]^−^	C_27_H_30_O_15_	2.2	285.0408
Daidzin	21.32	461.1108 [M + HCOOH-H]^−^	C_21_H_2_0O_9_	5.2	415.1031 [M-H]^−^, 253.0523
Genistin	28.30	477.1048 [M + HCOOH-H]^−^	C_22_H_22_O_12_	3.1	431.0973 [M-H]^−^, 269.0474
Daidzein	39.57	253.0523 [M-H]^−^	C_15_H_10_O_4_	8.7	225.0543, 224.0486, 209.0623, 197.0588, 91.0185
Genistein	49.61	269.0474 [M-H]^−^	C_15_H_10_O_5_	−4.8	241.0490, 225.0543, 224.0486, 201.0550, 159.0453, 133.0300
Coumestrol	49.76	267.0286 [M-H]^−^	C_15_H_8_O_5_	−2.6	167.0480

^a^ Determined by high-energy function of MS^e^ method.

**Table 2 nutrients-14-03822-t002:** Inhibitory activity of the major constituents of PL-L-70E against hMAOs.

Compounds	hMAO-A	hMAO-B	SI ^a^
	IC_50_ (Mean ± SD, µM)		
Robinin	>400	>400	-
Quercetin-3-*O*-robinobioside	>400	>400	-
Rutin	387.12 ± 4.63	>400	-
Kaempferol-3-*O*-robinobioside	>400	>400	-
Nicotiflorine	>400	>400	-
Daidzin	>400	>400	-
Genistin	>400	>400	-
Daidzein	304.05 ± 4.72	356.86 ± 1.05	0.85
Genistein	4.77 ± 0.51	3.42 ± 0.39	1.39
Coumestrol	1.99 ± 0.68	77.79 ± 2.10	0.02
Selegiline hydrochloride ^b^	12.57 ± 0.51	0.38 ± 0.001	33.08

^a^ The selective index (SI) = hMAO-A IC_50_/hMAO-B IC_50._ ^b^ Positive control.

**Table 3 nutrients-14-03822-t003:** Enzyme kinetic property and binding mode of coumestrol on MAO-A.

Compounds	Inhibition Mode	Inhibition Constant (*K_i_*, µM)	Binding Energy (kcal/mol) ^a^	H-Bond Interaction Residues ^b^	Other Interaction Residues ^b^
Coumestrol	Competitive	1.32	−9.36	FAD, Tyr444	FAD and Phe352 (π–π T shaped), Tyr407 and Tyr444 (π–π stacking), Ile335, and Leu337 (π–alkyl)
Harmine ^c^	ND	ND	−8.43	ND	Tyr407 (π-π stacking, π-Alkyl), FAD (Van der Waals), Cys323 (π –sulfur), Ile335 (π–σ, π–alkyl), Tyr444, Ile180, and Leu337 (π–alkyl)

^a^ Determined by Autodock 4.2. ^b^ Determined by Discovery studio v17.2. ^c^ Co-ligand of human MAO-A (2z5x) obtained from Protein Data Bank. ND Not detected.

**Table 4 nutrients-14-03822-t004:** Pharmacokinetic properties of coumestrol.

Model Name	Predicted Values
Log P_o/w_ ^a^	2.94
BBB penetration ^b^	0.76
HIA ^c^	93.51
P-glycoprotein	Non-inhibition
Carcino-rat/mouse	Negative
CNS MPO score ^d^	5.24

^a^ The ratio of 1-octanol to water in the log of the solvent partitioning coefficient. ^b^ Absorption levels below 0.1 are considered low, 0.1 to 2.0 are considered moderate, and above 2.0 are considered good. ^c^ Absorption levels of 0 to 20% are considered low, 20 to 70% are considered moderate, and 70 to 100% are considered good. ^d^ CNS multiparameter optimization (MPO), >4.0: desirability score.

## Data Availability

Not applicable.

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
