# Peer review of "Phytoestrogen Coumestrol Selectively Inhibits Monoamine Oxidase-A and Amyloid β Self-Aggregation"

_nutrients, 2022, doi:10.3390/nu14183822_

Round 1

Reviewer 1 Report

The current work focuses on Phytoestrogen coumestrol selectively inhibits monoamine oxi- 2 dase-A and amyloid β self-aggregation. The experimental work appears to have been carried out well. However, a few points deserve attention for further publication. I suggest that it is accepted for publication after the following revisions:

- ABSTRACT: Phytoestrogen coumestrol selectively inhibits monoamine oxi- 2 dase-A and amyloid β self-aggregation: What parameters were optimized? Authors must include numbers with the results found. Stability of the biocatalyst? How much enzyme were utilized to process? Furthermore, what are the conditions of enzymatic reactions? Temperature, pH, ionic strength, for example. This information should be included in the abstract.

- INTRODUCTION:

- In this study, Phytoestrogen coumestrol selectively inhibits monoamine oxi- 2 dase-A and amyloid β self-aggregation: physical and covalent were implied for enzyme preparation? What the advantages? Additionally, the spacer arm, the steric hindrances for the enzyme- reaction caused by this groups when compared to the others groups? These strategies used should be better explained in the manuscript.

- Monoamine oxidase (MAO) are very special enzymes, having a peculiar mechanism of action. This information must be clear in the introduction to present manuscript.

- A paragraph describing the properties, application, mechanism of actuation to Monoamine oxidase (MAO) must be included in the manuscript.

- What is the origin of this Monoamine oxidase (MAO)? Is it a commercial ? Genetically modified? How was it produced?

- Phytoestrogen coumestrol selectively inhibits monoamine oxi- 2 dase-A and amyloid β self-aggregation:: Improved biocatalytic, kinetic, and efficient increased the activity? This process needs to be explained in the introduction of the manuscript.

- Phytoestrogen coumestrol selectively inhibits monoamine oxi- 2 dase-A and amyloid β self-aggregation: What optimization strategy was used? Why was it used? This information needs to be explained in the introduction of the manuscript.

- In this study, Phytoestrogen coumestrol selectively inhibits monoamine oxi- 2 dase-A and amyloid β self-aggregation: Enzymes were purified? The conventional methods for purification are ultrafiltration, precipitation and affinity chromatography. However some of these methods are complicated, laborious, time-consuming and expensive. This information must be clear in the introduction.

- The contribution and importance of these studies in the work performed must be explained in the introduction of the manuscript.

MATERIALS:

- Include the concentration of solutions to enzyme.

METHODS:

- Include the molar concentration of all the chemicals used, the way the methods are presented, not possible reproducibility.

- Preparation procedure: Please include more details, temperature, amount of protein per gram, pH, molar ratio, ionic strength.

- RESULTS AND DISCUSSION:

- The influence of substrate systems to monoamine oxi- 2 dase-A and amyloid β: The influence of temperature on the monoamine oxi- 2 dase-A and amyloid β stability was also investigated at various temperatures? The monoamine oxi- 2 dase-A and amyloid β showed how about stability than the free enzyme? The thermal stability Enzyme prepared is one of the most important application criteria for diferent applications. This stability depends on the enzyme preparation strategy. It also depends on the stabilization of the enzyme. This discussion could be improved. Please include in the manuscript.

- The stability monoamine oxi- 2 dase-A and amyloid β: in organic solvents, metal ions, or detergent enables its wide application in synthesis processes which nowadays are in great demand from the point of view of industrial enzymes. The effect of organic solvents on the β-xylosidase BH3683 was used to prepare FeSO 4 -CE-MOF-BH3683 microparticles:  activity was studied? For example, in the presence of ethanol, methanol, dimethyl sulfoxide (DMSO), dioxane, n-hexane, tert-butanol, acetone or 2-propanol?

- Monoamine oxi- 2 dase-A and amyloid β: Comparison of the free enzyme in terms of kinetic parameters: Authors need to compare these results with other results in the literature.

 - Was determined the full loading of enzyme prepared under the optimized conditions? This information must be clear in the manuscript.

 - The free enzyme may experience protein aggregation (mainly near to the isoelectric point). This may be caused by undesired enzyme- interactions where inactivation that can stabilize incorrect enzyme structures. This results must be cleared with the following manuscript: DOI: 10.1039/C6RA03627A. RSC Adv. 2016, 6, 27329–27334.This information must be clear in the manuscript.

 - The optimization of enzyme preparation process, the preparations shown having diffusion limitations? Considering the strategy presented in this manuscript. Please, this should be explained in the manuscript. What were the optimum conditions?

 - Effect of solution pH since the solution pH affects the generation of hydroxyl radicals and also influences the surfasse charge and interface potential properties of the catalyst, it is one of the important factors. Rhodanese as the biocatalyst showed considerable improves in the kinetic parameters in terms of activity, specific activity, Km and Vmax, optimum pH and Temperature?

 - Reusability of monoamine oxi- 2 dase-A and amyloid β:  The reusability of enzyme particles is very important while considering enzymatic reactions. Enzyme reusability was accounted for continuous application of the enzyme?

 - Monoamine oxi- 2 dase-A and amyloid β: Other factors that cause the loss of durability and stability of the biocatalysts should be explained in the manuscript.

 - Please, check all references according to the author's instructions.

- Include more details in the figures (error bars) and tables captions.

- The manuscript must be formatted according to the journal's 

Author Response

Response to Reviewer 1

The current work focuses on Phytoestrogen coumestrol selectively inhibits monoamine oxi- 2 dase-A and amyloid β self-aggregation. The experimental work appears to have been carried out well. However, a few points deserve attention for further publication. I suggest that it is accepted for publication after the following revisions:

- ABSTRACT: Phytoestrogen coumestrol selectively inhibits monoamine oxi- 2 dase-A and amyloid β self-aggregation: What parameters were optimized? Authors must include numbers with the results found. Stability of the biocatalyst? How much enzyme were utilized to process? Furthermore, what are the conditions of enzymatic reactions? Temperature, pH, ionic strength, for example. This information should be included in the abstract.

Answer : According to reviewer’s comment, we have revised the abstract. Please check the revised manuscript.

- INTRODUCTION:

- In this study, Phytoestrogen coumestrol selectively inhibits monoamine oxi- 2 dase-A and amyloid β self-aggregation: physical and covalent were implied for enzyme preparation? What the advantages? Additionally, the spacer arm, the steric hindrances for the enzyme- reaction caused by this groups when compared to the others groups? These strategies used should be better explained in the manuscript.

Answer : We used commercially available human MAO isozymes. And, we have included the additional description of the importance of the in-silico docking experiment in understanding the steric hindrance caused by the inhibitor’s functional group. Please check the revised manuscript.

- Monoamine oxidase (MAO) are very special enzymes, having a peculiar mechanism of action. This information must be clear in the introduction to present manuscript.

- A paragraph describing the properties, application, mechanism of actuation to Monoamine oxidase (MAO) must be included in the manuscript.

Answer : We have described the properties, application, mechanism of action of MAO in the introduction section. Please check the revised manuscript.

- What is the origin of this Monoamine oxidase (MAO)? Is it a commercial ? Genetically modified? How was it produced?

Answer : We used commercially available human MAO isozymes as described in method section (2.1.). 

- Phytoestrogen coumestrol selectively inhibits monoamine oxi- 2 dase-A and amyloid β self-aggregation:: Improved biocatalytic, kinetic, and efficient increased the activity? This process needs to be explained in the introduction of the manuscript.

Answer : We found active inhibitor from Pueraria lobata leaves through an enzyme-substrate reaction assay. And, enzyme-kinetic experiments were conducted according to the change in the concentration of substrate and inhibitor, and as a result, it was found that coumestrol, an active inhibitor, is a competitive inhibitor. A series of processes for this experiment are described at the end of the introduction, so please check the revised manuscript.

- Phytoestrogen coumestrol selectively inhibits monoamine oxi- 2 dase-A and amyloid β self-aggregation: What optimization strategy was used? Why was it used? This information needs to be explained in the introduction of the manuscript.

Answer: Please check the last part of introduction. We have described the strategy to discover the active compound from Pueraria lobata.

- In this study, Phytoestrogen coumestrol selectively inhibits monoamine oxi- 2 dase-A and amyloid β self-aggregation: Enzymes were purified? The conventional methods for purification are ultrafiltration, precipitation and affinity chromatography. However some of these methods are complicated, laborious, time-consuming and expensive. This information must be clear in the introduction.

Answer : We used commercially available human MAO isozymes and included the amount of enzyme for assay. Please check the revised manuscript.

- The contribution and importance of these studies in the work performed must be explained in the introduction of the manuscript.

Answer : We have included the contribution and importance of these studies in the introduction section. Please check the revised the manuscript.

MATERIALS:

- Include the concentration of solutions to enzyme.

Answer : We have included the concentration of enzyme. Please check the “Sections 2.4 and 2.5”

METHODS:

- Include the molar concentration of all the chemicals used, the way the methods are presented, not possible reproducibility.

- Preparation procedure: Please include more details, temperature, amount of protein per gram, pH, molar ratio, ionic strength.

Answer : We have revised the method section. Please check the “Sections 2.4 and 2.5”.

- RESULTS AND DISCUSSION:

- The influence of substrate systems to monoamine oxi- 2 dase-A and amyloid β: The influence of temperature on the monoamine oxi- 2 dase-A and amyloid β stability was also investigated at various temperatures? The monoamine oxi- 2 dase-A and amyloid β showed how about stability than the free enzyme? The thermal stability Enzyme prepared is one of the most important application criteria for diferent applications. This stability depends on the enzyme preparation strategy. It also depends on the stabilization of the enzyme. This discussion could be improved. Please include in the manuscript.

Answer : We did not measure the stability of MAO and amyloid beta under different temperature. In the further studies, it is need to check the thermodynamic effect of the inhibitor on the enzyme through a scan of changes in thermal stability in the free enzyme and inhibitor-enzyme complex. We have added these issues to the discussion section. Please check the revised manuscript.

- The stability monoamine oxi- 2 dase-A and amyloid β: in organic solvents, metal ions, or detergent enables its wide application in synthesis processes which nowadays are in great demand from the point of view of industrial enzymes. The effect of organic solvents on the β-xylosidase BH3683 was used to prepare FeSO 4 -CE-MOF-BH3683 microparticles:  activity was studied? For example, in the presence of ethanol, methanol, dimethyl sulfoxide (DMSO), dioxane, n-hexane, tert-butanol, acetone or 2-propanol?

Answer : All test samples were diluted in reaction buffer containing 10% or less DMSO, and solvent-control group was tested together for each assay to minimize false positive/false negative effects. We have added these issues to the result section (3.2.). Please check the revised manuscript.

- Monoamine oxi- 2 dase-A and amyloid β: Comparison of the free enzyme in terms of kinetic parameters: Authors need to compare these results with other results in the literature.

Answer : We have included the comparison between these data and the previous data in the result section (3.3.). Please check the revised manuscript.

 - Was determined the full loading of enzyme prepared under the optimized conditions? This information must be clear in the manuscript.

Answer : We established the conditions for optimization of enzymatic reactions and conditions for aggregation of amyloid beta peptides. We have added these issues to the result section (3.2. and 3.4.). Please check the revised manuscript.

 - The free enzyme may experience protein aggregation (mainly near to the isoelectric point). This may be caused by undesired enzyme- interactions where inactivation that can stabilize incorrect enzyme structures. This results must be cleared with the following manuscript: DOI: 10.1039/C6RA03627A. RSC Adv. 2016, 6, 27329–27334.This information must be clear in the manuscript.

Answer : As reviewer commented, free enzyme/peptide may experience protein aggregation. Thus, Amyloid beta peptide was pretreated with hexafluoroisopropanol (HFIP) to obtain a homogenous starting material in a non-amyloidogenic conformation. We have added this process in the method section (2.6.). Please check the revised manuscript.

 - The optimization of enzyme preparation process, the preparations shown having diffusion limitations? Considering the strategy presented in this manuscript. Please, this should be explained in the manuscript. What were the optimum conditions?

Answer : We established the conditions for optimization of enzymatic reactions and conditions for aggregation of amyloid beta peptides. And detailed conditions were described in method section. We have added these issues to the result section (3.2. and 3.4.). Please check the revised manuscript.

 - Reusability of monoamine oxi- 2 dase-A and amyloid β:  The reusability of enzyme particles is very important while considering enzymatic reactions. Enzyme reusability was accounted for continuous application of the enzyme?

Answer : We did not measure the reusability of MAO after treatment of coumestrol. Reversibility assay is needed to confirm whether coumestrol is a reversible inhibitor. We have added these issues to the discussion section. Please check the revised manuscript.

- Effect of solution pH since the solution pH affects the generation of hydroxyl radicals and also influences the surfasse charge and interface potential properties of the catalyst, it is one of the important factors. Rhodanese as the biocatalyst showed considerable improves in the kinetic parameters in terms of activity, specific activity, Km and Vmax, optimum pH and Temperature?

 - Monoamine oxi- 2 dase-A and amyloid β: Other factors that cause the loss of durability and stability of the biocatalysts should be explained in the manuscript.

Answer : We have discussed about the factors affecting MAO activity in the result section (3.2.). Please check the revised manuscript.

 - Please, check all references according to the author's instructions.

Answer : We have checked the references according to the journal’s format. Please check the revised manuscript.

- Include more details in the figures (error bars) and tables captions.

Answer : We have included more details in the captions of figures and tables.

- The manuscript must be formatted according to the journal's

Answer : We have revised the manuscript according to the journal’s format. Please check the revised manuscript.

In conclusion, we have revised the manuscript according to Reviewer 1’s suggestions, point by point. Kindly, review the revised manuscript and provide more constructive comments.

We appreciate the reviewer for thorough review of our paper and such a constructive comments and suggestions. We hope this revision meets the standards for publication.

Regards,

Authors

Reviewer 2 Report

Review of nutrients-1924691

This is a good pharmacokinetics paper intertwining with neurology applications for Alzheimer disease. However, there are some aspects (quantitative data, data processing, writing, etc.) to be corrected and added, as follows:

  1. All equations must be presented in this manuscript. For example, to obtain Figure 2B (the Lineweaver-Burk plot, or 1/[S] vs 1/V, as function of coumestrol concentration), there has to be a table that shows raw data such as [S], V, and coumestrol concentration, as well as processed data such as 1/[S] and 1/V.
  2. How to obtain Figure 2C, if there is no information about any equation related to the kinetics??? Add also the table containing the complete raw data to get Figure 2C.
  3. How to obtain Figure 2D if there is no information about any equation related to the kinetics??? Add also the table containing the complete raw data to get Figure 2D
  4. Section 2.8: What are the data needed or fed into the software and server? Please write the details in the Section 2.8.
  5. Figure 3A: Add the table containing the complete raw data to get Figure 3A.
  6. Figure 3B: Add the table containing the complete raw data to get Figure 3B. Add the table containing the complete raw data to get IC50 values of curcumin and coumestrol displayed in the legend of Figure 3B.
  7. Line 217-219: If these pharmacokinetic results came from references then please add appropriate references. If those came from Section 2.8, then what are the data needed or fed into the software and the server?
  8. Line 233: Change “middle” to be “moderate”. Note: the word “middle” refers to a “location”, while the right context in this sentence is about a “level”.
  9. Reference 2: Check the original source, https://doi.org/10.1016/j.ejmech.2020.112787 where “Anti-Alzheimer” is started with uppercase letters A and A, respectively.
  10. Reference 4: Check the original source, https://doi.org/10.1186/s13195-017-0279-1 where “oxidase B” is written with uppercase B.
  11. Reference 4: Check the original source, https://doi.org/10.1186/s13195-017-0279-1 where “Alzheimer” is written with uppercase A.
  12. Reference 6: Check the original source, https://doi.org/10.1016/j.biopha.2020.110734 where “Pueraria” is written in italic.
  13. Reference 7: Check the original source, https://doi.org/10.1016/j.jep.2016.10.007 where “Pueraria lobata” is written in italic, started with uppercase P.
  14. Reference 7: Check the original source, https://doi.org/10.1016/j.jep.2016.10.007 where “phospatase 1B” is written with uppercase B.
  15. Reference 8: Check the original source, https://doi.org/10.3389/fphar.2017.00599 where “Pueraria lobata” is written in italic, started with uppercase P.
  16. Reference 11: Check the original source, https://doi.org/10.1093/jb/118.5.974 where “oxidase A and B” is written with uppercase A, and B, respectively.
  17. Reference 13: Check the original source, https://doi.org/10.1021/cn3000982 where “Alzheimer” is written with uppercase A.
  18. Reference 16: Check the original source, https://doi.org/10.1155/2016/1423052  where “oxidase-A and -B” is written with uppercase A, and B, respectively.
  19. Reference 18: Check the original source https://doi.org/10.1021/cn100008c where “CNS MPO”, an abbreviation, is written in all uppercase letters.
  20. Reference 20: Check the original source https://doi.org/10.1038/s41598-018-38219-6 where “RNA” is written in all uppercase letters. I don’t have to explain what RNA is.
  21. Reference 21: Castro, C. C --> NOT Canal Castro, C. Check the original in this link: https://doi.org/10.1016/j.brainres.2012.07.025
  22. Reference 23: Check the original source, https://doi.org/10.1186/s12906-016-1525-y  where “Biochanin-A” is written with uppercase B, and A, respectively.
  23. Reference 23: Check the original source, https://doi.org/10.1186/s12906-016-1525-y where “oxidase B” is written with uppercase B.
  24. Reference 24: Check the original source, https://doi.org/10.3390/molecules25173896 where “oxidase-B” is written with uppercase B.
  25. Reference 24: Check the original source, https://doi.org/10.3390/molecules25173896 where “Glycyrrhiza uralensis” is written in italic, and started with uppercase G.
  26. Reference 25: Check the original source, https://doi.org/10.1021/jm060882y where “MAO-A and MAO-B” is written with all uppercase letters, because they are an abbreviation.
  27. Reference 26: Check the original source, https://doi.org/10.1074/jbc.m110920200 where “oxidase A and B” is written with uppercase A, and B, respectively.
  28. Reference 28: Check the original source, https://doi.org/10.3390/molecules23040785  where “Pueraria lobata” is written in italic, and started with uppercase P.
  29. Reference 28: Check the original source, https://doi.org/10.3390/molecules23040785 where “Alzheimer” is started with uppercase letter A.
  30. Reference 30: Check the original source, https://doi.org/10.4061/2011/690121 where “MDR-1” is written with all uppercase letters.
  31. Reference 30: Check the original source, https://doi.org/10.4061/2011/690121 where “Abcb1” is started with uppercase A.

Author Response

Response to Reviewer 2

This is a good pharmacokinetics paper intertwining with neurology applications for Alzheimer disease. However, there are some aspects (quantitative data, data processing, writing, etc.) to be corrected and added, as follows:

All equations must be presented in this manuscript. For example, to obtain Figure 2B (the Lineweaver-Burk plot, or 1/[S] vs 1/V, as function of coumestrol concentration), there has to be a table that shows raw data such as [S], V, and coumestrol concentration, as well as processed data such as 1/[S] and 1/V.

Answer : We have added all equations for kinetic results in the method section (2.5.). And, raw data for Figure 2B was tabulated in Table S1. Please check the revised manuscript and supplementary files.

How to obtain Figure 2C, if there is no information about any equation related to the kinetics??? Add also the table containing the complete raw data to get Figure 2C.

How to obtain Figure 2D if there is no information about any equation related to the kinetics??? Add also the table containing the complete raw data to get Figure 2D

Answer : Figure 2C and 2D were automatically generated using exploratory EK macro of SigmaPlot 12.0 software. And, the secondary plots were generated using the V values (= RLU/60min) obtained from kinetic assay as data set (Table S1).

We have added all equations for kinetic results in the method section (2.5.). And, raw data for Figure 2C and 2D was tabulated in Table S1. Please check the revised manuscript and supplementary files.

Section 2.8: What are the data needed or fed into the software and server? Please write the details in the Section 2.8.

Answer : We have added the details in the section 2.8. Please check the revised manuscript.

Figure 3A: Add the table containing the complete raw data to get Figure 3A.

Figure 3B: Add the table containing the complete raw data to get Figure 3B. Add the table containing the complete raw data to get IC50 values of curcumin and coumestrol displayed in the legend of Figure 3B.

Answer : Raw data for Figures 3A and 3B was tabulated in Table S2. Please check the supplementary files.

Line 217-219: If these pharmacokinetic results came from references then please add appropriate references. If those came from Section 2.8, then what are the data needed or fed into the software and the server?

Answer : Pharmacokinetic results were predicted based on the chemical structure of coumestrol using two softwares (MarvinSketch and PreADMET).  We have added more details in the result section (3.5.).

Line 233: Change “middle” to be “moderate”. Note: the word “middle” refers to a “location”, while the right context in this sentence is about a “level”.

Answer : We have revised it.

Reference 2: Check the original source, https://doi.org/10.1016/j.ejmech.2020.112787 where “Anti-Alzheimer” is started with uppercase letters A and A, respectively.

Reference 4: Check the original source, https://doi.org/10.1186/s13195-017-0279-1 where “oxidase B” is written with uppercase B.

Reference 4: Check the original source, https://doi.org/10.1186/s13195-017-0279-1 where “Alzheimer” is written with uppercase A.

Reference 6: Check the original source, https://doi.org/10.1016/j.biopha.2020.110734 where “Pueraria” is written in italic.

Reference 7: Check the original source, https://doi.org/10.1016/j.jep.2016.10.007 where “Pueraria lobata” is written in italic, started with uppercase P.

Reference 7: Check the original source, https://doi.org/10.1016/j.jep.2016.10.007 where “phospatase 1B” is written with uppercase B.

Reference 8: Check the original source, https://doi.org/10.3389/fphar.2017.00599 where “Pueraria lobata” is written in italic, started with uppercase P.

Reference 11: Check the original source, https://doi.org/10.1093/jb/118.5.974 where “oxidase A and B” is written with uppercase A, and B, respectively.

Reference 13: Check the original source, https://doi.org/10.1021/cn3000982 where “Alzheimer” is written with uppercase A.

Reference 16: Check the original source, https://doi.org/10.1155/2016/1423052  where “oxidase-A and -B” is written with uppercase A, and B, respectively.

Reference 18: Check the original source https://doi.org/10.1021/cn100008c where “CNS MPO”, an abbreviation, is written in all uppercase letters.

Reference 20: Check the original source https://doi.org/10.1038/s41598-018-38219-6 where “RNA” is written in all uppercase letters. I don’t have to explain what RNA is.

Reference 21: Castro, C. C --> NOT Canal Castro, C. Check the original in this link: https://doi.org/10.1016/j.brainres.2012.07.025

Reference 23: Check the original source, https://doi.org/10.1186/s12906-016-1525-y  where “Biochanin-A” is written with uppercase B, and A, respectively.

Reference 23: Check the original source, https://doi.org/10.1186/s12906-016-1525-y where “oxidase B” is written with uppercase B.

Reference 24: Check the original source, https://doi.org/10.3390/molecules25173896 where “oxidase-B” is written with uppercase B.

Reference 24: Check the original source, https://doi.org/10.3390/molecules25173896 where “Glycyrrhiza uralensis” is written in italic, and started with uppercase G.

Reference 25: Check the original source, https://doi.org/10.1021/jm060882y where “MAO-A and MAO-B” is written with all uppercase letters, because they are an abbreviation.

Reference 26: Check the original source, https://doi.org/10.1074/jbc.m110920200 where “oxidase A and B” is written with uppercase A, and B, respectively.

Reference 28: Check the original source, https://doi.org/10.3390/molecules23040785  where “Pueraria lobata” is written in italic, and started with uppercase P.

Reference 28: Check the original source, https://doi.org/10.3390/molecules23040785 where “Alzheimer” is started with uppercase letter A.

Reference 30: Check the original source, https://doi.org/10.4061/2011/690121 where “MDR-1” is written with all uppercase letters.

Reference 30: Check the original source, https://doi.org/10.4061/2011/690121 where “Abcb1” is started with uppercase A.

Answer : We have revised that. Please check the revised manuscript. Thank you for your detailed comments.

In conclusion, we have revised the manuscript according to Reviewer 2’s suggestions, point by point. Kindly, review the revised manuscript and provide more constructive comments.

We appreciate the reviewer for thorough review of our paper and such a constructive comments and suggestions. We hope this revision meets the standards for publication.

Regards,

Authors

Round 2

Reviewer 2 Report

Review of nutrients-1924691-v2 The authors have addressed well the issues previously raised. This manuscript can be accepted now. Note: Line 121-124: Please make the font type and font size to be uniform with the rest of the manuscript. This correction can be performed during the proofreading stage. Thanks.